# Associations between Indoor Air Pollution and Acute Respiratory Infections among Under-Five Children in Afghanistan: Do SES and Sex Matter?

**DOI:** 10.3390/ijerph16162910

**Published:** 2019-08-14

**Authors:** Juwel Rana, Jalal Uddin, Richard Peltier, Youssef Oulhote

**Affiliations:** 1Department of Biostatistics and Epidemiology, School of Public Health and Health Sciences, University of Massachusetts, Amherst, MA 01003, USA; 2Department of Environmental and Occupational Health Sciences, EHESP French School of Public Health, Paris 93210, France; 3South Asia Institute for Social Transformation (SAIST), Dhaka 1205, Bangladesh; 4Department of Epidemiology, University of Alabama at Birmingham, Birmingham, AL 35233, USA; 5Department of Environmental Health, School of Public Health and Health Sciences, University of Massachusetts, Amherst, MA 01003, USA

**Keywords:** socioeconomic status, indoor air pollution, acute respiratory infection, cooking fuel, under-five children

## Abstract

*Background*: Low-income families often depend on fuels such as wood, coal, and animal dung for cooking. Such solid fuels are highly polluting and are a primary source of indoor air pollutants (IAP). We examined the association between solid fuel use (SFU) and acute respiratory infection (ARI) among under-five children in Afghanistan and the extent to which this association varies by socioeconomic status (SES) and gender. *Materials and Methods*: This is a cross-sectional study based on de-identified data from Afghanistan’s first standard Demographic and Health Survey (DHS) conducted in 2015. The sample consists of ever-married mothers with under-five children in the household (*n* = 27,565). We used mixed-effect Poisson regression models with robust error variance accounting for clustering to examine the associations between SFU and ARI among under-five children after adjusting for potential confounders. We also investigated potential effect modification by SES and sex. Additional analyses were conducted using an augmented measure of the exposure to IAP accounting for both SFU and the location of cooking/kitchen (High Exposure, Moderate, and No Exposure). *Results*: Around 70.2% of households reported SFU, whereas the prevalence of ARI was 17.6%. The prevalence of ARI was higher in children living in households with SFU compared to children living in households with no SFU (adjusted prevalence ratio (aPR) = 1.10; 95% CI: (0.98, 1.23)). We did not observe any effect modification by SES or child sex. When using the augmented measure of exposure incorporating the kitchen’s location, children highly exposed to IAP had a higher prevalence of ARI compared to unexposed children (aPR = 1.17; 95% CI: (1.03, 1.32)). SES modified this association with the strongest associations observed among children from the middle wealth quintile. *Conclusion*: The findings have significant policy implications and suggest that ARI risk in children may be reduced by ensuring there are clean cookstoves as well as clean fuels and acting on the socio-environmental pathways.

## 1. Introduction

Globally, an estimated 5.6 million under-five children died in 2016. Preventable causes such as diarrhea, malaria, undernutrition, and respiratory infections account for the majority of all deaths among under-five children in developing countries [1]. One in every five Afghan children dies from Acute Respiratory Infection (ARI) before reaching their fifth birthday [2,3]. Evidence suggests that multiple risk factors such as indoor air pollution (IAP) and social determinants including socioeconomic status (SES) and neighborhood are responsible for ARI related deaths in resource-poor settings [4,5,6,7].

A growing body of literature suggests that exposure to such toxic pollutants is causally linked to a host of morbidities such as ARI, cardiovascular, and pulmonary conditions among young children [8,9,10,11,12]. Smoke from biomass and coal contains a large number of toxins and particulate matters of different sizes such as nitrogen dioxide, carbon monoxide, methylene chloride, and dioxins [9,10,11]. Exposure to these pollutants is unusually high among women and under-five children because they tend to spend more time in proximity to fires while cooking and heating [13]. Studies have shown that pollution generated in kitchens and heating areas immediately spreads into living areas, which is how children and other household members are exposed to IAP [13,14].

Families with low SES are often confronted with living in environmentally hazardous and disadvantaged neighborhoods. Low-income and occupational constraints force many families to live in substandard housing or maintain living conditions which expose residents and children to an array of toxins, pollutants, and infections [15,16,17]. For instance, low-income families often depend on the use of low-cost but high-pollution biofuels such as wood, coal, straws, and animal dung as the primary source of energy for cooking and heating in many developing countries due to limited availability and access to non-solid fuels such as electricity and natural gas [8,18,19]. Moreover, most of the impoverished families living in rural areas and particularly those engaged in agriculture find it easier and cheaper to use biomass fuels collected from their surroundings and farmlands [5,18].

The IAP related under-five mortality and associated morbidity disproportionately affects children from poorer households. Socioeconomic gradient in child mortality and morbidity is well-established in that children with poorer SES fare worse in health across countries, every region of the world, and between population groups in individual countries [20,21,22]. For instance, under-five mortality rates were double in households in the lowest wealth quintile compared to households in the highest wealth quintile in Afghanistan, in 2015 [23]. In explaining this association between SES and mortality, Link and Phelan propose that individuals with greater access to flexible resources are better positioned with greater means to avoid disease risks and achieve health goals [24].

This paper aims to examine the association between SFU and ARI among under-five children. To our best knowledge, there is no previous study that examined the association between exposure to IAP and important child morbidity, such as ARI, using a nationally representative sample from Afghanistan. Using the first-ever standard Demographic and Health Survey (DHS) of 2015 conducted in Afghanistan, we examined this association and determine whether the exposure to IAP is a risk factor for ARI among under-five children, and the extent to which SES and sex may modify this association.

## 2. Materials and Methods

### 2.1. Study Design and Setting

This was a cross-sectional study based on the de-identified data from Afghanistan’s first standard Demographic and Health Survey (AfDHS) conducted in 2015. This nationally representative household survey was conducted among ever-married women aged 15–49 years. AfDHS followed a two-stage stratified sampling design including both urban and rural areas in each of the 34 provinces of Afghanistan from June 2015 to February 2016 using MEASURE DHS model questionnaires (https://dhsprogram.com/Data/). From 24,395 households, all eligible ever-married women of reproductive age 15–49 with 31,063 children of under five years old were interviewed with a response rate of 98 percent. This sample was limited to the last birth happening within three years of the survey. After missing cases and incomplete information were deleted, the analytic sample consisted of 27,565 ever-married women with under-five children, for whom the complete information on all variables of interest was available for both children and women.

### 2.2. Ethical Approval and Consent to Participate 

The survey protocol for the primary data collection was approved by the ICF Institutional Review Board and the Ministry of Public Health of Afghanistan. We obtained the de-identified data from the DHS online archive. This is a public-use dataset. Informed consent was taken from each participant before the enrollment. 

### 2.3. Measure of ARI Outcome

ARI was the outcome of interest in this study. DHS asked all mothers whether their children suffered from cough in the two weeks prior to the survey. Mothers who answered in the affirmative were then asked whether the cough was compounded by short or rapid breathing problems in the two weeks preceding the survey. The outcome variable ARI, therefore, refers to the conditions in which a child suffered a cough with shortness of breathing or difficulty in breathing. We created a binary variable for ARI outcome assigning a value of 1 if the responses for both questions were ‘yes’ (e.g., suffered a cough and had short/rapid breathing and/or difficult breathing) and two other responses were coded 0 (e.g., no cough and no trouble breathing and had cough but no trouble breathing) [8,25,26].

### 2.4. Measures of Exposure to IAP

Exposure to IAP was measured using reporting of SFU. SFU was categorized into two groups: non-SFU (e.g., electricity, liquid petroleum gas, natural gas, biogas) and SFU (e.g., kerosene, coal, lignite, charcoal, wood, animal dung, straw/shrubs/grass). To improve the measure of exposure to IAP, we constructed an additional indicator of exposure to IAP accounting for both SFU and the location of cooking/kitchen. Children were categorized as highly exposed if the households used solid fuels and kitchen was inside the house (coded 2); moderately exposed if the households used solid fuels and the kitchen was in a separate building or outdoor (coded 1), and otherwise unexposed to IAP (coded 0).

### 2.5. Covariates and Potential Confounders

Previous literature found a significant relationship with multiple factors of ARI which have guided the selection of covariates for this study [21,22,23,24,25]. All variables included in the models were chosen *a priori*: child sex, child age in years, child size at birth, maternal age at birth, maternal education, parental occupation, household wealth quintiles, urbanity, mother’s smoking status, breastfeeding status, region, and season. Maternal age at birth was categorized: ≤24 years/25–35 years/35+ years. The ever-breastfeeding variable was binary in nature, which indicates whether the mother has ever breastfed their children or not (never/ever). Similarly, the mother’s smoking status was a binary variable (never/ever). The urbanity was categorized into rural and urban.

SES was measured by household wealth quintile, paternal and maternal education, and occupation. Household wealth index is a composite measure of a household’s cumulative wealth and living standard. In the AfDHS, the wealth index was constructed based on household’s ownership of selected assets such as television, bicycle, car, and dwelling characteristics, such as the source of drinking water, sanitation facilities, types of cooking fuels, and flooring construction materials. Using principal component analysis, each of these assets was assigned scores and based on the overall score; individual households were placed on a continuous scale of relative wealth. For this study, a new wealth index was reconstructed, excluding the types of cooking fuels as this was the main exposure of interest. Finally, all households were grouped into five wealth quintiles [27,28,29].

Level of parents’ education was measured with four categories: no education, primary, secondary, and above secondary. Mother’s occupation was categorized into five groups: not working, professionals, clerical/services, skill manual, and unskilled manual. Father’s occupation was categorized into five groups: professional, clerical/services, agriculture, skilled manual, and unskilled manual. Finally, the region variable was created by regrouping 34 provinces into eight regions: Capitals, Northeastern, Northern, Western, Central Highland, Eastern, Southern, and Southeastern. “Season” variable was created based on the months of interview: summer (June–August), autumn (September–November) and winter (December–February). The season, variable, was categorized into three: summer, autumn, and winter.

The final analyses included child age, child sex, maternal age at birth, maternal education, parental occupation, household wealth quintile, season, mother’s smoking status, breastfeeding status, and region as covariates.

However, father’s education and urbanity were excluded from the robust multivariable model due to multicollinearity with wealth quintile and region. Mother’s reported size of child at birth, a proxy of birthweight, was excluded because it may be in the causal pathway.

### 2.6. Data Analysis

All analyses took into account the sampling design to account for the clustering nature of the data and the probability of selection and non-response in the AfDHS. We first present the characteristics of the study population by ARI status and by exposure variables: solid fuel use and exposure to IAP. The association between solid fuel use and ARI outcome was examined by mixed-effect Poisson regression models, which accounted for survey design and adjusted for potential confounders cited above. We used mixed-effect Poisson regression with robust error variance to avoid inflation of estimates resulting from logistic regression when the outcome is common (>10%). Results were presented as adjusted Prevalence Ratios (aPRs). All statistical tests were two-sided, and a *p*-value < 0.05 was the level of significance. We used STATA 15 version (Stata Corp LP, College Station, TX, USA) for all statistical analyses and data management. 

AfDHS data were obtained from the MEASURES DHS. The data is available on the website of the global DHS program, which could be accessed after getting registered with the MEASURES DHS. Details: https://dhsprogram.com/data/Using-DataSets-for-Analysis.cfm.

## 3. Results

### 3.1. Sample Characteristics by Children’s ARI

The sample distribution and univariate association with different characteristics are presented in Table 1 and Table 2. The total of 27,565 under-five children was included in this study. Of all the under-five children, 51.7% were boys, and 48.3% were girls. Of the 27,565 children, 61.8% of children were average size at birth, and 63.0% were ever breastfed. The mean maternal age was 29 years old. Maternal education was very poor: 83.5% of all mothers had no education, and 87.8% of mothers were not involved in any paid work. The under-five children were predominantly from rural areas (77.3%), which reflects the place of residence where most of the Afghans live. Of all the children, about 41.6% of children were from poor households (poorest and poorer together). Of 27,565 children, 59.0% of children were from households with an indoor kitchen.

Of these sample characteristics, SFU, and exposure to IAP were the main exposures of interest (Table 1). The main predictors of SFU were SES indicators such as parental education, parental employment, and household wealth quintile along with urbanity, and geographical region (Table 2).

Overall, the prevalence of SFU was 70.2% in Afghanistan. Of the 27,565 under-five children, 17.6% suffered from ARI in the previous two weeks. The percentage of ARI was higher among children (18.7%) living in households using solid fuels than children (15.2%) from households using non-solid fuels. About 40.2% of children were highly exposed to IAP, 30% moderately exposed, and 29.8% unexposed to IAP. The percentage of ARI was higher among highly exposed children (20.8%) and moderately exposed children (15.8%), compared to unexposed children (15.2%).

### 3.2. Multivariable Associations between SFU and ARI

The mixed-effect Poisson regression was conducted to estimate the association between ARI and SFU after adjusting for child age, child sex, maternal age at birth, maternal education, parental occupation, household wealth quintiles, season, mother’s smoking status, breastfeeding, and region. Results show that the risk of ARI was significantly higher (unadjusted PR 1.19; 95% CI: (1.08, 1.32); *p* = 0.001) among children living in the households who reported solid fuel use compared to the children living in the households who reported non-solid fuel use. However, adjustment for confounders attenuated the association between solid fuel use and ARI (aPR 1.10; 95% CI: (0.98, 1.23); *p* = 0.11). We did not observe any effect modification in this association by SES and child sex (Figure 1 and Figure 2).

Additional analyses were conducted using an augmented measure of the exposure to IAP accounting for both SFU and the location of cooking (High Exposure, Moderate, and No Exposure). When using the augmented measure of exposure incorporating the kitchen’s location, children highly exposed to IAP (aPR 1.17; 95% CI: (1.03, 1.32)) had higher prevalence of ARI compared to unexposed children to IAP. SES modified this association with the strongest associations observed among children from the middle wealth quintile (Figure 2).

## 4. Discussion

Although previous studies suggest that socioeconomic inequalities are persistently associated with child morbidity and mortality, very few previous studies attempted to explain whether exposure to indoor air pollution a risk factor is also specifically in the context of Afghanistan. The objective of this study was to address this gap by looking at the associations of SFU and exposure to IAP with ARI. The combination of kitchen location with the SFU has improved the measure of exposures to IAP. Thus, this study used two measures of exposures-SFU and exposure to IAP combining SFU and kitchen location, which is a more inclusive measure of exposure to IAP and could reduce bias due to measurement error. The study further aimed to assess the effect modification of SES and gender on these associations.

The results suggest that children who were highly exposed to IAP were more likely to suffer from ARI compared to unexposed children. A similar association is also reported in other studies from developing countries [5,8,11,26]. Past studies routinely indicate that the combustion of solid fuels emits many toxic pollutants and young children quickly get exposed to these toxic pollutants and in turn, suffer from adverse health outcomes [9,10,12,14,25,30,31].

Cooking inside the house increases the concentrations of particulate matter in the household. Younger children tend to spend extended time with their mother in the house, near indoor fires and heating during cooking [10,26,32,33]. It is usually the case that young children are often carried on the back or placed near their mothers to sleep while they cook in the kitchen [8,34]. In particular, younger children become more susceptible to IAP and its health outcomes than their older counterparts because of their underdeveloped epithelial linings of the lungs [14]. Also, younger children have higher respiration, narrower airways, and larger lung surface, due to which they breathe 50% more polluted air under normal conditions than older children [33]. Moreover, the weak immune system is also associated with their vulnerability of IAP, and it is ARI outcome while children’s immune systems develop to resist against different infections with the increased age [14,35,36].

The analysis demonstrated that SES, as indicated by wealth quintiles, significantly modified the association between exposure to IAP and children’s ARI. Interestingly, this association was modified by SES with the strongest associations observed among children from the middle wealth quintile. Probably, children from poorest and poor wealth quintile were less exposed to IAP compared to children from the middle wealth quintile because poor households in low and middle-income countries (LMICs) tend to not cook much compared to people from middle wealth quintiles. Moreover, poor households may not have a specific cooking place and often cook in open places that ensure natural ventilation, hence the exposure to IAP in these households may be blunted by the open spaces even with the high prevalence of SFU. Conversely, households in the middle wealth quintile may have a specific cooking place inside the house and cook more than poor households, which increases exposures to IAP. However, previous studies showed that children from impoverished households are more likely to be exposed to IAP and health-threatening toxins stemming from the use of solid cooking fuels and there is a protective effect of increased household wealth on child health outcomes in the LMICs [7,18,26,32,34,35,37].

In this study, although generally girls and women spend more time indoors and are exposed to higher levels of air pollutants as a result, we did not find evidence of an effect modification by sex of the associations between IAP and ARI.

### Limitations

We acknowledge several limitations of our analysis. First, the analysis is based on cross-sectional data that does not allow us to infer the causal effects of the predictors on ARI. Second, the outcome measure of ARI is based on self-reporting of the mothers. Self-reported data are often erroneous due to long recall periods. However, it is highly unlikely that mothers would misreport the illness of their children when mothers were asked to recall the child’s ARI related symptoms within the two weeks before the survey. Third, regarding indoor air pollution, we used two proxy measures, such as the types of fuels used for cooking and the location of the kitchen [15,20,21]. Many may argue that these questions do not adequately measure the air quality and the extent of indoor air pollution in the household and the extent to which children get exposed to this polluted air. A better measure of child’s exposure to indoor air pollution is to measure concentrations of pollutants and how much time children spend in proximity to kitchen or heating areas where high pollution fuels burned for cooking and heating purposes. Additionally, there should be questions such as whether the household has enough ventilation and if there are better protective tools in the kitchen, such as a chimney in the kitchen stove so that smoke can escape. Future studies and surveys may add such questions to determine the exposure to and severity of indoor air pollution.

## 5. Conclusions

This research demonstrates that the higher exposure to IAP such as having a kitchen inside the house and solid fuel use appear to be significant risk factors for children’s ARI. Furthermore, the risk of ARI is modified by SES. The results suggest that children from the highly exposed group and households in the middle wealth quintile are in a most disadvantaged position to have a higher burden of ARI. Policymakers should pay attention to these social inequalities in ARI and pay particular attention to alleviating the risk of ARI by acting on the socio-environmental pathways through which child health is determined. In particular, a short-term intervention effort may focus on ensuring that more impoverished households get easier access to clean cooking stoves and household logistics that help build pollution-free housing.

## Figures and Tables

**Figure 1 ijerph-16-02910-f001:**
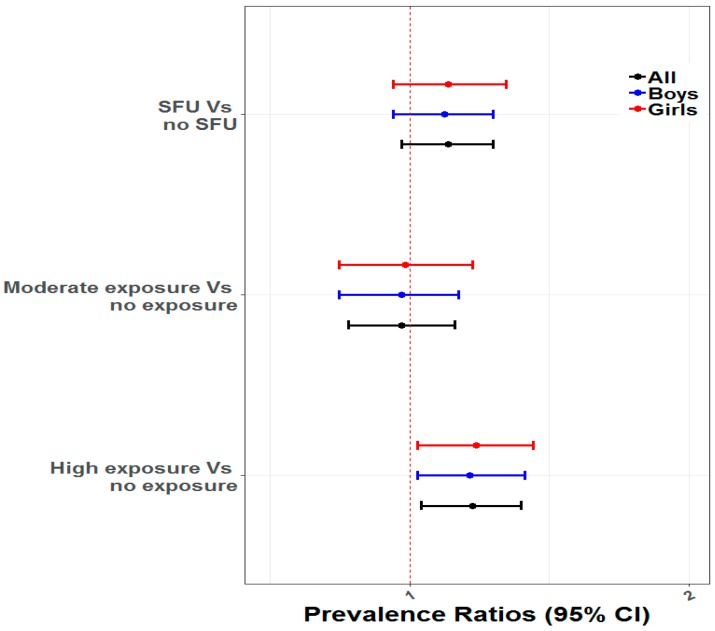
Associations of ARI with SFU (SFU and No SFU) and Exposure to IAP (No Exposure, Moderate Exposure and High Exposure) modified by sex of child.

**Figure 2 ijerph-16-02910-f002:**
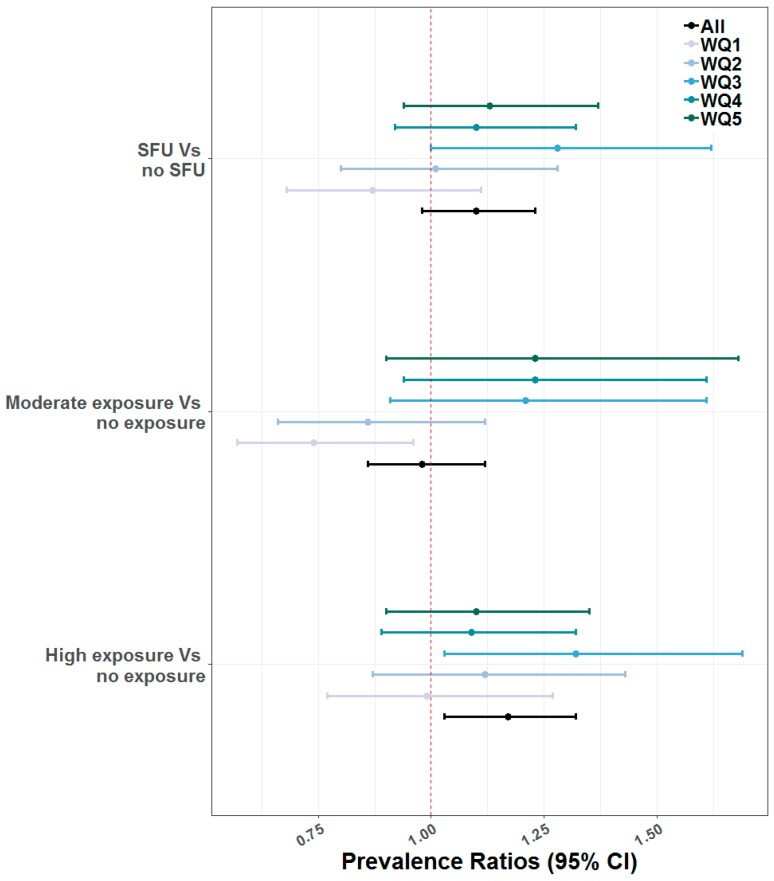
Associations of ARI with SFU (SFU and No SFU) and Exposure to IAP (No Exposure, Moderate Exposure and High Exposure) modified by Wealth Quintiles (WQs).

**Table 1 ijerph-16-02910-t001:** The weighted percentage of sample characteristics by Acute Respiratory Infection (ARI) among under-five children in Afghanistan.

Acute Respiratory Infection (ARI)			
Characteristics	Sample Distribution (%) *	ARI (%) *	Chi-Square Test *p*-Value
Overall	27,565 (100)	17.6	-
*Sex of the children*			
Boy	14,275 (51.7)	18.2	0.12
Girl	13,290 (48.3)	17.1	
*Size of child at birth*			
Smaller than average	3936 (14. 4)	18.2	<0.001
Average	16,878 (61.8)	16.5	
More than average	5731 (23.8)	21.6	
*Ever Breastfed*			
No	10,220 (37.0)	18.3	<0.001
Yes	17,221 (63.0)	17.3	
*Children’s age (in years)*			
0	5263 (19.2)	17.3	<0.001
1	5102 (18.4)	19.0	
2	5951 (21.8)	20.3	
3	5761 (21.0)	16.4	
4	5488 (19.6)	15.1	
*Mother’s age at birth (in years)*			
15–24 years	12,077 (44.5)	16.5	0.007
25–35 years	12,379 (44.3)	19.0	
36–49 years	3109 (11.2)	16.8	
*Mother’s education*			
No education	23,651 (83.5)	18.1	0.08
Primary	1889 (8.1)	17.2	
Secondary	1640 (7.0)	13.6	
Higher	385 (1.4)	12.7	
*Mother’s Occupation*			
Not working	24,085 (87.8)	17.2	0.08
Professional	1224 (6.7)	24.5	
Agriculture/self-employed	1370 (1.8)	16.9	
Skilled manual	440 (2.7)	16.9	
Unskilled manual	429 (0.9)	14.1	
*Mother’s smoking status*			
Non-smoker	26,470 (95.9)	17.5	0.03
Smoker	1095 (4.1)	22.0	
*Father’s education*			
No education	15,681 (57.6)	17.7	0.54
Primary	3804 (14.9)	18.0	
Secondary	6187 (21.3)	18.0	
Higher	1893 (6.2)	15.2	
*Father’s Occupation*			
Professional	5762 (17.6)	16.2	0.007
Clerical/Services	5068 (18.1)	15.6	
Agriculture/self-employed	7909 (29.6)	20.6	
Skilled manual	4112 (17.1)	16.0	
Unskilled manual	4714 (17.7)	17.9	
*Wealth quintile*			
Poorest	5802 (21.4)	22.9	<0.001
Poorer	5860 (20.2)	18.2	
Middle	5825 (19.0)	16.6	
Richer	5850 (20.6)	13.4	
Richest	4228 (18.8)	16.6	
*Urbanity*			
Urban	6590 (22.7)	17.6	0.95
Rural	20,975 (77.3)	17.7	
*Region*			
North Eastern	2682 (12.5)	19.4	<0.001
Northern	3417 (17.6)	15.1	
Western	3818 (15.7)	29.7	
Central Highland	1187 (2.1)	10.2	
Capitals	4016 (17.6)	11.5	
Eastern	4346 (8.7)	20.1	
Southern	4428 (17.3)	18.5	
South Eastern	3671 (8.5)	8.6	
*Season*			
Summer	4786 (20.1)	13.8	<0.001
Autumn	14,890 (49.8)	16.1	
Winter	7889 (30.1)	22.8	
*Location of Cooking*			
Indoor Kitchen	17,645 (59.0)	19.1	<0.001
Outdoor Kitchen	9920 (41.0)	15.5	
*Solid Fuel Use (SFU)*			
Unexposed	6799 (29.8)	15.2	0.02
Exposed	20,766 (70.2)	18.7	
*Exposure to IAP*			
Unexposed	6799 (29.8)	15.2	<0.001
Moderately exposed	7622 (30.0)	15.8	
Highly exposed	13,144 (40.2)	20.8	

* Sample size (*n*) are unweighted, and percentage of sample distribution and prevalence of ARI based on weighted analysis.

**Table 2 ijerph-16-02910-t002:** The weighted percentage of sample characteristics by Solid Fuel Use (SFU) and exposure to indoor air pollutants (IAP) (*n* = 27,565).

Characteristics	Solid Fuel Use (SFU)		Exposure to IAP			
	Prevalence of SFU *	*p*-Value	Unexposed *	Moderately Exposed *	Highly Exposed *	*p*-Value
Overall	70.2	-	29.8	30.0	40.2	-
*Sex of the children*						
Boy	69.7	0.31	30.3	30.0	39.7	0.46
Girl	70.8		29.3	30.0	40.7	
*Size of child at birth*						
Smaller than average	67.2	0.01	32.8	31.1	36.0	0.005
Average	71.5		28.5	30.7	40.8	
More than average	66.9		33.1	26.6	40.3	
*Ever breastfed*						
No	68.7	0.007	31.4	29.4	39.2	0.01
Yes	71.1		28.9	30.4	40.8	
*Children’s age (in years)*						
0	68.6	0.09	31.4	28.8	39.9	0.10
1	68.8		31.3	31.2	37.6	
2	70.4		29.6	29.9	40.5	
3	70.2		29.8	29.6	40.6	
4	72.9		27.1	30.8	42.1	
*Mother’s age at birth (in years)*						
15–24 years	69.5	0.03	30.5	29.1	40.4	0.03
25–35 years	69.7		30.4	29.8	39.8	
36–49 years	75.2		24.8	34.5	40.7	
*Mother’s education*						
No education	75.5	<0.0001	24.5	31.5	44.0	<0.001
Primary	49.1		50.9	22.9	26.2	
Secondary	40.2		59.8	23.3	16.9	
Higher	25.2		74.9	15.6	9.6	
*Mother’s occupation*						
Not working	70.3	<0.0001	29.7	30.1	40.2	<0.001
Professional	57.7		42.3	19.3	38.4	
Agriculture/self-employed	99.8		0.2	42.2	57.5	
Skilled manual	76.5		23.5	47.7	28.8	
Unskilled manual	76.0		24.0	24.7	51.3	
*Mother’s smoking status*						
Non-smoker	69.7	0.002	30.3	29.8	39.8	0.002
Smoker	82.9		17.1	35.1	47.9	
*Father’s education*						
No education	78.2	<0.0001	21.8	34.4	43.8	<0.001
Primary	63.6		36.5	27.1	36.4	
Secondary	60.4		39.6	24.1	36.3	
Higher	46.1		54.0	17.1	28.9	
*Father’s occupation*						
Professional	69.9	<0.0001	30.1	31.0	38.9	<0.001
Clerical/Services	50.6		49.4	19.8	30.9	
Agriculture/self-employed	91.2		8.8	36.0	55.2	
Skilled manual	52.4		47.6	26.0	26.4	
Unskilled manual	72.6		27.4	33.5	39.2	
*Wealth quintile*						
Poorest	94.7	<0.0001	5.3	51.0	43.7	<0.001
Poorer	88.0		12.0	37.7	50.4	
Middle	76.9		23.1	25.3	51.6	
Richer	58.1		41.9	19.6	38.5	
Richest	29.6		70.4	14.2	15.5	
*Urbanity*						
Urban	19.3	<0.0001	80.7	9.0	10.3	<0.001
Rural	85.1		14.9	36.2	48.9	
*Region*						
North Eastern	85.7	<0.0001	14.3	62.8	22.9	<0.001
Northern	67.7		32.3	38.6	29.1	
Western	80.3		19.7	40.3	40.0	
Central Highland	92.6		7.4	29.1	63.5	
Capitals	37.7		62.3	18.9	18.9	
Eastern	85.1		14.9	6.4	78.6	
Southern	69.2		30.8	10.6	58.7	
South Eastern	82.2		17.8	32.2	50.1	
*Season*						
Summer	22.8	<0.0001	77.2	10.3	12.5	<0.001
Autumn	76.4		23.6	35.2	41.2	
Winter	91.7		8.3	34.7	57.0	

* Percentages of SFU and exposure to IAP are based on the weighted analysis.

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
