# Peer review of "Associations between Indoor Air Pollution and Acute Respiratory Infections among Under-Five Children in Afghanistan: Do SES and Sex Matter?"

_ijerph, 2019, doi:10.3390/ijerph16162910_

Round 1

Reviewer 1 Report

General - report percentages to the same number of decimal places throughout.

General -  report p-values to the same number of decimal places (or significant figures if preferred) throughout.

P2 line 5 - define ARI on first use.

P2 line 26 - change affect to affects

P3 line 3 - can you explain what this means, as it implies study children were under 3 yet elsewhere and in the table children are aged 0 to 4.

P3 line 11 - remove issues

P3 lines 25 to 32 - remove repetition.

P4 line 2 - were parental occupations added to the model(s) separately or did some recoding occur to give a composite for parental occupation?

P4 line 23 - how is size at birth operationalised?

Author Response

Response to Reviewer 1 Comments

General - report percentages to the same number of decimal places throughout.

Authors’ Response: Thank you for the comment. We have checked and reported the remaining percentages to the same number of decimal places throughout the manuscript.

General - report p-values to the same number of decimal places (or significant figures if preferred) throughout.

Authors’ Response: We have checked and reported the remaining p-values to the same number of decimal places throughout the manuscript.

P2 line 5 - define ARI on first use.

Authors’ Response: We have defined ARI on first use in the abstract- P1 lines 18-19. However, we understood that we have to define all abbreviations on first use from the introduction as well. Thus, we have defined ARI in P2 line 5.

P2 line 26 - change affect to affects

Authors’ Response: We have changed 'affect' to 'affects' in P2 line 27.

P3 line 3 - can you explain what this means, as it implies study children were under 3 yet elsewhere and in the table, children are aged 0 to 4.

Authors’ Response: We think you mentioned “under 5” instead of under 3 above. “Current age of child” is also an original AfDHS variable used in our study. In the table, 0-4 means: 0=less than 12 months, 1=12 months to less than 24 months, 2=24 months to less than 36 months, 3=36 months to less than 48 months and 4=48 months to 59 months. 

P3 line 11 - remove issues

Authors’ Response: removed “issues” and include “..in..” before breathing in P3 line 10.

P3 lines 25 to 32 - remove repetition.

Authors’ Response: We have removed the repetition of potential covariates in P3 lines 27-29.

P4 line 2 - were parental occupations added to the model(s) separately or did some recoding occur to give a composite for parental occupation?

Authors’ Response: Parental occupations (mother’s occupation and father’s occupation) were both included in the models separately.

P4 line 23 - how is size at birth operationalised?

Authors’ Response: “Size at birth” is mothers’ estimates of their infant’s size at birth. It is an original variable of AfDHS.

According to the final report of AfDHS 2015, “written records or mothers’ reports of birth weight were available for only 14% of live births in the 5 years before the survey. Due to the low percentage of births for which birth weight was available, it is unlikely that this figure is representative of all births in the country.” Hence, AfDHS “collected information on mothers’ estimates of their infant’s size at birth”. AfDHS said that “although the mother’s estimate of size is subjective, it can be used as a useful proxy for the child’s weight” (Central Statistics Organization, Ministry of Public Health, and ICF. 2017, 159).

There were five categories originally: very large, larger than average, average smaller than average and very small. Due to lack of sufficient samples for each category, we have regrouped these into three: Smaller than average, average and more than average.

Reviewer 2 Report

General comments:

The article has good structure and be well organized. The figures and Tables have also well expressed the results well accurately. However, there are still many shortcomings here to be improved before being published.

Firstly, the measurements of ARI outcome and exposure to IAP are not accurate enough. Secondly, the irrelevant contents in the table seem to be too much when the emphases in the discussion are just SES and gender. Finally, the whole article ignores the impact of gender in the associations between indoor air pollution and acute respiratory infections, and this doesn’t match the title. 

Some specific comments:

1. Page 3 line 5: The measurement of AIR outcome is too rough

2. Page 3 line 20: Solid fuel usage and kitchen location can’t represent personal exposure accurately.  

3. Page 3 line 48: What are the principles when regrouping 34 provinces?

4. Page 3 line 48: How to categorize season in detail?

5. Page 4 line 4: Do mother's education and urbanity have multicollinearity with wealth quintile and region?

6. Page 4 line 5: Has the causal pathway between the size of child at birth and birthweight been confirmed or analyzed?

7. Page 4 line 21: The format of digital writing is not uniform with previous (eg. Page 2 line 49 and page 3 line 3).

8. Page 9 line 14: School-age children may spend much time outdoors and school, so the IAP in the house can’t represent children’s exposure.

9. Page 10 line 5: How the sex influences the association between IAP and ARI in detail when we can find that there is no obvious difference between boys and girls according to figure 1?

10. Page 11 line 17: There may be a formatting error here.

11. Table 1: How to explain the ratio of ARI in 2-year-old-children is higher than that in 1 and 0-year-old-children when children's immune systems develop to resist against different infections with an increased age?

12. Table 1: How to explain that the ratio of ARI in urban and rural are nearly the same when the AIR is usually different?

13. Table 2 & Figure 2: How to explain that the wealth quintile’s influence in moderately and high exposure are different?  

Author Response

Response to Reviewer 2 Comments

General comments:

The article has good structure and be well organized. The figures and Tables have also well expressed the results well accurately. However, there are still many shortcomings here to be improved before being published.

Firstly, the measurements of ARI outcome and exposure to IAP are not accurate enough. Secondly, the irrelevant contents in the table seem to be too much when the emphases in the discussion are just SES and gender. Finally, the whole article ignores the impact of gender in the associations between indoor air pollution and acute respiratory infections, and this doesn’t match the title. 

Authors’ Response: Thank you for your critical remarks for the improvement of our manuscript.

Firstly, ARI outcome and IAP exposure measures are well-established in the prevailing literature. However, we have also acknowledged the limitations of using the measurement of ARI outcome and IAP exposures in P9 lines 45-51 and P10 lines 1-4.

Secondly, Table 1 and 2 present all descriptive statistics along with the univariate associations of all potential covariates with both exposures and outcome. We agree that Table 2 may seem less important contents, but we believe that it helps researcher and reader to understand the association between exposures of interest and all potential covariates; thus, understanding what are the main potential confounders and covariates to include in the models. It also helps to understand the study easily and have comprehensive idea about the outcome and exposures. Hence, we have pressed on the association between IAP and ARI along with the effect modification of SES and gender in the discussion, which were our questions of interest and relevant to our argument. As in all epidemiological studies, the causal question of interest is the one discussed and for which the models are built, as all the other covariates cannot be interpreted in the same way.  

Finally, we agree with the reviewer that we did not discuss more on the impact of gender in the discussion, but we have stated in the manuscript that we have tested the effect modification of sex and already mentioned the result in P1 line 31 and P7 lines 10-11. We now added the following sentences in the discussion (P9 lines 40-42) for further clarification:

"In this study, although generally girls and women spend more time indoors and are exposed to higher levels of air pollutants as a result, we did not find evidence of an effect modification by sex of the association between IAP and ARI."

Some specific comments:

Page 3 line 5: The measurement of AIR outcome is too rough

Authors’ Response: The measurement of ARI outcome is well-established in the literature and DHS program uses this definition for understanding ARI situation among under-five children for 90 developing countries in the world (DHS program). Hence, we believe that the measurement of ARI outcome is justified given the large dataset we have. In such large population-based surveys, it is generally agreed to have easier and cheaper measurements of health outcomes that can provide a general picture of health indicators. It would have been easier to measure ARI more precisely in small cohort studies. We have also acknowledged the limitations of using the measurement of ARI outcome in P9 lines 45-49.  

Page 3 line 20: Solid fuel usage and kitchen location can’t represent personal exposure accurately.

Authors’ Response: We agree with the reviewer that solid fuel use (SFU) and kitchen location are not the accurate measure of personal exposure to indoor air pollution (IAP). However, SFU is also the well-established proxy measure of exposure to IAP.

Furthermore, "the combination of kitchen location with the SFU has improved the measure of exposures to IAP. Thus, this study used two measures of exposures-SFU and exposure to IAP combining SFU and kitchen location, which is more inclusive measure of exposure to IAP and could reduce bias due to measurement error" newly included in P9 lines 6-9 for clarification. 

Furthermore, we have acknowledged the limitations of using the proxy measures SFU and IAP exposures in P9 lines 49-51 and P10 lines 1-4. 

Page 3 line 48: What are the principles when regrouping 34 provinces?

Authors’ Response: The principle was to adjust for regional/geographical variation of IAP and ARI in Afghanistan as well as to ensure better interpretability. In the original dataset, there was no variable named “region or geographical location”. Hence, we have created the variable “region” by regrouping 34 provinces according to the nationally and internationally recognized geographic regions in Afghanistan. Regrouping these also allows a better adjustment in the models.

Page 3 line 48: How to categorize season in detail?

Authors’ Response: There are four seasons in Afghanistan and season is an important covariate to measure the association between IAP and ARI because exposure to IAP varies a lot in terms of seasons, and also ARI. In the data set, there was no specific variable “season” but the date of the interview was available. Thus, “Season variable was created based on the months of the interview: summer (June-August), autumn (September-November) and winter (December-February)” newly included in the method section in P3 line 48 and P4 line 1.  

Page 4 line 4: Do mother's education and urbanity have multicollinearity with wealth quintile and region?

Authors’ Response: Yes, mother’s education and urbanity also have multicollinearity with wealth quintile and region. But, not to the point where it creates computational issues, thus, all the models converged correctly, and we did not see signs of multicollinearity problems such as inverse signs, large CIs, and high variance inflation factors.

Page 4 line 5: Has the causal pathway between the size of child at birth and birthweight been confirmed or analyzed?

Authors’ Response: Formally we did not analyze the causal association between the size of child at birth and birth weight because: firstly, we did not include birth weight in this study. In P4 line 5, we have mentioned that size of child at birth was used as the proxy of birth weight because AfDHS found only written records of birth weight for 14% under-five child only and suggested to use “size of child at birth” as proxy of birth weight (Central Statistics Organization, Ministry of Public Health, and ICF. 2017, 159).

Secondly, we meant that the size of child at birth, proxy of birth weight, could be in the causal pathway of IAP and ARI, which has been discussed in the following literature. Adjusting for an intermediate variable such as this one would have biased our estimates towards the null.

VanderWeele TJ, Mumford SL, Schisterman EF. Conditioning on intermediates in perinatal epidemiology. Epidemiology. 2012; 23:1–9.

Wilcox AJ, Weinberg CR, Basso O. On the pitfalls of adjusting for gestational age at birth. Am J Epidemiol. 2011;174: 1062–8.

Hernandez-Diaz S, Schisterman EF, Hernan MA. The birth weight “paradox” uncovered? Am J Epidemiol. 2006;164: 1115–20.

Page 4 line 21: The format of digital writing is not uniform with previous (eg. Page 2 line 49 and page 3 line 3).

Authors’ Response: We would like to have more details about this comment as we were unable to locate the issues.

Page 9 line 14: School-age children may spend much time outdoors and school, so the IAP in the house can’t represent children’s exposure.

Authors’ Response: Indeed, which requires measurements inside and outside the house that are not available in our study, nor in all the representative studies of DHS. We have supported our argument with previous studies. However, we also believe that our research was focused on under-five children only and the large majority in Afghanistan are not school-age children. In South Asia, formal education or schooling starts at the age of six years old.

Page 10 line 5: How the sex influences the association between IAP and ARI in detail when we can find that there is no obvious difference between boys and girls according to figure 1?

Authors’ Response: Thank you very much for bringing this mistake to our attention. It was not an intentional mistake and we have removed “and sex” in P10 line 11.

Page 11 line 17: There may be a formatting error here.

Authors Response: We have changed the format by converting all words into lowercase except the very first word of the title: “….Socioeconomic status and health: the potential role of environmental risk exposures..” in P11 lines 21-22.

Table 1: How to explain the ratio of ARI in 2-year-old-children is higher than that in 1 and 0-year-old-children when children's immune systems develop to resist against different infections with an increased age?

Authors’ Response: Thank you for raising this point. Table 1 represents the descriptive statistics of ARI and associated covariates. Hence, we cannot conclude about the association between ARI and child age. It could appear due to lots of factors such as confounders. For example, the adjusted multivariable models showed that an increased age was the protective factor of ARI for children.  

Table 1: How to explain that the ratio of ARI in urban and rural are nearly the same when the AIR is usually different?

Authors’ Response: The ratio of ARI in urban and rural are almost similar which as congruence with the AfDHS final report. The plausible explanation could be that Afghanistan does not have many highly urbanized areas in terms of infrastructural and socio-economic development. Such as, electricity is unavailable in most of the regions both in the city and countryside. In the urban area, disadvantaged people who do not have access to clean fuel are also dependent on solid fuels and live with poverty. Moreover, crisis-affected areas also have poor air quality that may reduce these rural-urban differences in AIR quality remarkably.

Table 2 & Figure 2: How to explain that the wealth quintile’s influence in moderately and high exposure are different?

Authors Response: The influence of wealth quintile on moderately and high exposure almost similar in Table 2 and Figure 2. Table 2 clearly depicted that the moderate and high exposure to IAP varies in poorest vs richer and richest wealth quintile. However, the high exposure to IAP was lower among poorest children compared to children with poorer and middle wealth quintile, that has also been reflected in the effect medication of wealth quintile- Figure 2. The plausible explanation regarding the effect of wealth quantile on moderately and high exposure to IAP are in P9 lines 29-35. We speculate that in the poorest quintiles, although the use of SFU may be higher, but the effects will also depend on how much cooking is done at home, and therefore on the availability of food. If availability of food is lower in the poorest quintiles, there might be lower cooking activities, and therefore lower exposures to IAP even if the prevalence of SFU is high.